# Corn Cropping System and Nitrogen Fertilizers Technologies Affect Ammonia Volatilization in Brazilian Tropical Soils

César Santos [1], Sheila Isabel do Carmo Pinto [2], Douglas Guelfi [1,*], Sara Dantas Rosa [3], Adrianne Braga da Fonseca [1], Tales Jesus Fernandes [4], Renato Avelar Ferreira [2], Leandro Barbosa Satil [2], Ana Paula Pereira Nunes [1] and Konrad Passos e Silva [2]

1    Laboratory of Fertilizers Technologies—INNOVA FERT, Department of Soil Science, Federal University of Lavras—UFLA, P.O. Box 3037, Lavras 37203-202, MG, Brazil
2    Department of Agricultural Sciences, Federal Institute of Minas Gerais, Campus Bambuí, Bambuí 38900-000, MG, Brazil
3    Faculty of Agronomy and Veterinary Medicine, University of Brasilia, Brasilia 70910-900, DF, Brazil
4    Department of Statistics, Federal University of Lavras, Lavras 37203-202, MG, Brazil
*    Correspondence: douglasguelfi@ufla.br

**Abstract:** The adoption of technologies for N fertilization has become essential for increasing the N use efficiency in no-till (NT) systems in Brazil. Thus, this study aimed to quantify ammonia losses, N removal in grains, and second crop season yield in no-till and conventional (T) areas that received the application of different N fertilizers and their technologies. Ammonia volatilization, N extraction in grains, and corn yield in response to the application of conventional fertilizers were compared to urea treated with urease inhibitors in NT and conventional systems. The treatments were: no-N (Control); Prilled urea (PU); urea + N-(n-Butyl) thiophosphoric triamide ($U_{NBPT}$); urea + Cu + B ($U_{CuB}$); ammonium nitrate (AN), and ammonium sulfate (AS). In the NT system, the $N-NH_3$ losses were 49% higher than in the conventional; without differences in corn yield. The fertilizers AN and AS had the lowest $N-NH_3$ losses, regardless of the tillage system. $U_{NBPT}$ reduced the mean $N-NH_3$ loss by 33% compared to PU. $U_{NBPT}$ (1200 mg kg$^{-1}$) and $U_{NBPT}$ (180 mg kg$^{-1}$) reduced the $N-NH_3$ losses by 72% and 22%, respectively, compared to PU in the NT system. We noticed that the NBPT concentration to be used in soils under NT should be adjusted, and a reduction of $N-NH_3$ losses does not directly reflect an increase in yield and N extraction by corn.

**Keywords:** no-till; urea technologies; nitrogen use efficiency; urease inhibitors

## 1. Introduction

Among the nutrients most used in the fertilization of corn, nitrogen (N) stands out as it is required in large amounts. It is estimated that 286 kg N are required for a corn yield of 12 Mg ha$^{-1}$ [1], which is the average yield of corn in highly productive areas in Brazil. Urea is the most used source of N-fertilizer in corn production systems in Brazil, with values corresponding to 60% of the N used in Brazilian agriculture. However, the agronomic efficiency of conventional urea applied without soil incorporation is low owing to the losses of $N-NH_3$ by volatilization with negative impacts to the environment [2–4]. Aligned with worldwide trends, initiatives or guidelines related to the mitigation of $N-NH_3$ losses from N-fertilizers, such as conventional urea, may also increase in Brazil [5,6].

The $N-NH_3$ losses increase with the use of urea without any kind of technologies and are intensified by soil and climate conditions such as pH, CEC, humidity, temperature, and levels of urease activity in the soil, relative air humidity, rainfall, and presence of crop residues on the soil [7–9]. Considering the typical tropical conditions in Brazil, these losses can be worsened, with average losses of $N-NH_3$ around 30% in varying cultivation systems [4,10,11].

Besides the favorable conditions of the tropical climate, the increase in the number of areas cultivated with corn under no-till (NT) is another key factor that favors the N-$NH_3$ losses. The areas under NT have increased in Brazil due to the NT advantages, such as increased organic matter content, reduced losses of soil and nutrients by erosion, and increased maintenance of soil moisture [12,13]. On the other side, the presence of straw on the soil surface in NT systems intensifies the N-$NH_3$ losses, particularly with the application of urea [14], which may cause N-$NH_3$ losses 25% greater than in conventional tillage systems [3]. The increased content of soil organic matter enhances the activity of urease, which is the enzyme that operates in the hydrolysis of urea into N-$NH_3$ and $CO_2$ [14,15]. Moreover, the straw prevents the direct contact between the fertilizer and the soil, reducing its incorporation by the rainwater [2,7].

Ammonia volatilization reduces the N retention in soils, soil fertility, grain yield, and N use efficiency, and in some world regions, it may lead to environment pollution, with direct impacts in the production costs and social [16]. Thereby, in the last years, the search for technologies that reduce N losses from urea has expanded. One manner to prevent N-$NH_3$ losses consists of the mechanical incorporation of urea to the soil [3,9,17] or by the rainwater and irrigation [2]. However, the mechanical incorporation is not recommended in the NT system due to the presence of straw and the fact that soil disruption should be avoided. Moreover, mechanical incorporation is rarely adopted by Brazilian farmers, even in conventional systems. Thus, the use of conventional urea without incorporation is an option that is becoming uncommon in agricultural regions of the world, in which there are initiatives and guidelines to mitigate ammonia emissions [5].

Considering this challenge, the fertilizer industry and researchers around the world have turned their attention to the production of slow-release, controlled released, or stabilized fertilizers to improve the N use efficiency in agriculture [3,14].

Stabilized fertilizers have additives that delay or inhibit some transformation process of N in the soil, such as the urease activity or the nitrification reaction. Several compounds such as NBPT (N-(n-Butyl) thiophosphoric triamide), NPPT (N-(n-Propyl) thiophosphoric triamide), metallic cations, boron, and organic N compounds have been studied as an alternative to reduce urease activity in the soil and minimize N losses to the atmosphere [2,5,18,19].

Thus, it is crucial to adopt fertilizer technologies that are able to ensure greater N use efficiency towards the 4R's stewardship. The use of slow-release, controlled, and stabilized N fertilizers in order to mitigate N losses in corn production systems is very relevant within agronomic and environmental scenarios.

Thus, the hypotheses of this study were: (1) the fertilizers: ammonium nitrate (AN), ammonium sulfate (AS), urea treated with NBPT ($U_{NBPT}$), and urea treated with Cu and B ($U_{CuB}$) reduce N-$NH_3$ losses compared to conventional urea (PU) under no-till and conventional systems; (2) N-$NH_3$ losses from conventional urea are higher in the no-till system, but urea technologies or the use of ammonium nitrate and ammonium sulfate can reduce such losses. To test these hypotheses, the present study was performed in two cropping systems (conventional and no-till) for two years under field conditions in southeastern Brazil. The rationale of this study was to evaluate technologies for N-fertilizers and their efficiency to mitigate N-$NH_3$ losses and improve corn nutrition and yield in the second crop season.

## 2. Material and Methods

### 2.1. Preparation and Characterization of the Used Fertilizers

In the 2017/2018 crop, the N sources were: (1) Urea treated with NBPT, with 46% N and 180 mg NBPT $kg^{-1}$ ($U_{NBPT}$); (2) Prilled Urea, with 46% N (PU); (3) Urea treated with Cu and B, with 43% N, 0.3% Cu, and 0.3% B ($U_{CuB}$); (4) Ammonium nitrate, with 33% N (AN), and (5) Ammonium sulfate, with 19% N and 22% S (AS). All fertilizers were purchased from a fertilizer store. As for the 2018/2019 crop season, the $U_{NBPT}$ was treated in the laboratory since the NBPT concentration in the fertilizer obtained in the 2017/2018

season was lower than that described in the commercial fertilizer (530 mg kg$^{-1}$). The other fertilizers were obtained from a fertilizer store.

The treatment of urea with NBPT used in the 2018/2019 crop season was performed at the Laboratory of Technologies for Fertilizers at the Universidade Federal Lavras. For that, a solution including diethanolamine (CAS number 111-42-2) (70%) and NBPT (30%) was prepared. From this solution, 8.6 g were taken and homogenized with 2 kg of granular urea in a bench top mixer. Then, the NBPT concentration was determined by high-performance liquid chromatography (HPLC), model HP1100 Agilent with diode-array detection (DAD) [20], which was 1200 mg kg$^{-1}$.

### 2.2. Site Description and Management Practices

Two experiments with corn (*Zea mays*), hybrid 2B-512PW of the Dowscience® company were performed during the second crop season of 2017/2018 and 2018/2019, after the cultivation of soybean (*Glycine max*), in Medeiros and Bambuí, Minas Gerais state, Brazil (20°07′00″ S, 46°09′55″ W and 20°06′47″ S, 46°10′00″ W, respectively (Figure 1).

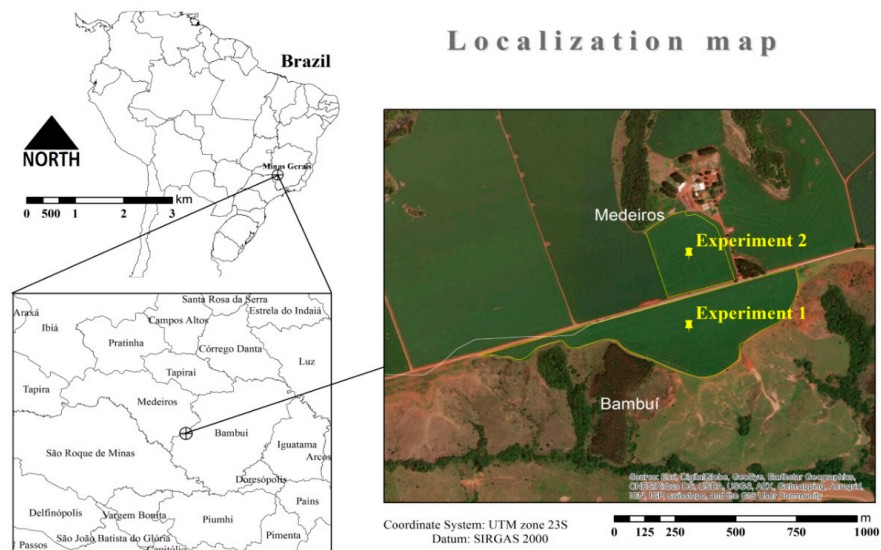

**Figure 1.** Location of the experimental areas, Experiment 1 (crop season 2017/2018) and Experiment 2 (crop season 2018/2019).

The experiments were installed in a slope within a hilly region, in a soil classified as Acrudox [21].

The information regarding the main characteristics of the sites and crop seasons are summarized on Table 1.

**Table 1.** Characteristics of the experiments performed in the 2017/2018 and 2018/2019 crop seasons.

| Characteristics | 2017/2018 Medeiros, Minas Gerais State, Brazil. | 2018/2019 Bambuí, Minas Gerais State, Brazil. |
|---|---|---|
| Soil type | Acrudox | Acrudox |
| Latitude | 20°07′00″ S | 20°06′47″ S |
| Longitude | 46°09′55″ W | 46°10′00″ W |
| Mean annual temperature (°C) | 20.3 | 21.3 |
| Mean annual precipitation (mm) | 1457 | 1369 |
| Accumulated precipitation (mm) [a] | 134.5 | 155.5 |
| Total N (kg ha$^{-1}$, 0–0.20 m) | 2330 (T); 2024 (NT) | 2250 (T); 1765 (NT) |
| NO$_3^-$ (kg ha$^{-1}$, 0–0.20 m) | 24.6 (T), 18.7 (NT) | 55.75 (T), 62 (NT) |
| NH$_4^+$ (kg ha$^{-1}$, 0–0.20 m) | 8.2 (T), 27.7 (NT) | 44 (T), 36.5 (NT) |
| pH (0–0.20 m) [b] | 5.5 (T), 5.6 9 (NT) | 5.9 (T), 5.8 (NT) |

[a] Accumulated after 29 days of fertilization [b] pH in water 1:2.5 (*v/v*).

### 2.3. Cropping Systems and Field Management

The rationale of this study emerged after reading some papers previously published in the scientific literature. Table 2 lists the main results on the topic found in the scientific literature.

**Table 2.** Ammonia (N-NH$_3$) losses in no-till (NT) and till (T) till systems.

| Crops | Fertilizers and N Rates | NT N-NH$_3$ Losses kg ha$^{-1}$ | T N-NH$_3$ Losses kg ha$^{-1}$ | References |
|---|---|---|---|---|
| Corn | Urea (60 kg N ha$^{-1}$) | 3 | 2.3 | [22] |
| Rice | Urea | 24.8 | 0.63 | [23] |
| | Coated urea Cu + B (120 kg N ha$^{-1}$) | 11.6 | 0.01 | |
| Corn 28 years | Urea (160 kg N ha$^{-1}$) | 12.7 | 2.1 | [15] |
| *Camelina sativa* L. 20 years | Urea | 0.51 | 0.51 | [24] |
| | Urea + NBPT (90 kg N ha$^{-1}$) | 0.28 | 0.29 | |
| Wheat/Wheat | Diammonium phosphate (80 kg N ha$^{-1}$) | 16.8 | 16 | [25] |
| Wheat/faba bean 20 years | Urea + ammonium nitrate (120 kg N ha$^{-1}$) | 10.4 | 10 | |
| Corn | Urea | 18 | - | [26] |
| | Urea + Cu + B (100 kg ha$^{-1}$) | 11 | | |
| Corn | Urea | 21.1 | - | [2] |
| | Urea + Cu + B (150 kg ha$^{-1}$) | 17.3 | | |
| Corn 20 years | Urea | 22.0 | - | [14] |
| | Urea + NBPT (200 kg ha$^{-1}$) | 4.4 | | |
| Corn 20 years | Urea | 26.0 | - | [4] |
| | Urea + NBPT (150 kg ha$^{-1}$) | 5.4 | | |

Since the aim of this study is the comparison between the N fertilizers and their technologies, and also the influence of the tillage systems on the N-NH$_3$ losses by volatilization, we decided to simulate the conventional tillage within a NT area that had approximately 15 years of implantation. For that, the straw was manually removed from the plots designed to represent the conventional tillage, and the soil was plowed up to 20 cm depth in the 2016/2017 and 2017/2018 summer crop seasons (Figure S1). Thus, the corn sowing (second crop season) was performed after soybean cultivation in the summer for both years. Before the sowing, soil samples were collected for chemical and physical characterization. Six composite samples were collected, obtained from a homogenous mixture of ten simple soil samples collected at the 0–0.05, 0.05–0.10, and 0.10–0.20 m soil depths. The clay, silt, and sand content values were 40, 31, 29% and 44; 36; 20% for the 2017/2018 and 2018/2019 crop seasons, respectively, and the results of the chemical analysis are presented on Table 3 and Table S1. The amounts of macro and micronutrients applied to the soil were based on the results of the soil analyses and calculated according to CFSEMG [27].

**Table 3.** Soil organic carbon and nitrogen contents and carbon and nitrogen stocks at different soil depths in conventional tillage (T) and no-till (NT) systems in the 2017/2018 and 2018/2019 crop seasons.

| Sist. | Depth | OC | TN | C/N | N-NH$_4^+$ | N-NO$_3$ | BD | E$_{N.T.}$ | E$_{C.O.}$ | E$_{NH4}^+$ | E$_{NO3}^-$ | E$_{NM}$ |
|---|---|---|---|---|---|---|---|---|---|---|---|---|
| | | | | | | 2017/2018 Crop Season | | | | | | |
| | cm | —g kg$^{-1}$— | | | —mg dm$^{-3}$— | | kg dm$^{-3}$ | | | —kg ha$^{-1}$— | | |
| T | 0–5 | 17 | 2.5 | 6.8 | 13.9 | 29.0 | 0.9 | 1125 | 7650 | 6.6 | 13.8 | 20.4 |
| | 5–10 | 18 | 3.3 | 5.4 | 32.3 | 92.6 | 1.1 | 1849 | 9900 | 18.1 | 51.8 | 70.0 |
| | 10–20 | 18 | 2.8 | 6.4 | 6.6 | 14.4 | 1.1 | 3139 | 19,800 | 3.8 | 16.4 | 20.2 |
| | 0–20 | 18 | 2.8 | 6.4 | 29.8 | 75.2 | —— | 2330 | 14,287 | 8.0 | 24.5 | 32.5 |
| NT | 0–5 | 25 | 2.8 | 9.0 | 45.1 | 38.9 | 1.2 | 1656 | 15,000 | 26.5 | 22.8 | 49.3 |
| | 5–10 | 22 | 2.1 | 10.4 | 50.8 | 13.6 | 1.2 | 1231 | 13,200 | 29.5 | 7.9 | 37.4 |
| | 10–20 | 19 | 2.3 | 8.2 | 50.1 | 6.1 | 1.1 | 2606 | 20,900 | 27.4 | 6.7 | 34.1 |
| | 0–20 | 21 | 2.4 | 9.0 | 98.1 | 32.3 | —— | 2024 | 17,500 | 27.7 | 11.0 | 38.7 |

**Table 3.** *Cont.*

| Sist. | Depth | OC | TN | C/N | N-NH$_4^+$ | N-NO$_3$ | BD | E$_{N.T.}$ | E$_{C.O.}$ | E$_{NH4}^+$ | E$_{NO3}^-$ | E$_{NM}$ |
|---|---|---|---|---|---|---|---|---|---|---|---|---|
| | | | | | | 2018/2019 Crop season | | | | | | |
| T | 0–5 | 14 | 3.9 | 3.6 | 40.9 | 39 | 1.0 | 1955 | 7000 | 21 | 19 | 40 |
| | 5–10 | 16 | 2.9 | 5.5 | 141 | 81 | 1.0 | 1465 | 8000 | 69 | 40 | 109 |
| | 10–20 | 14 | 2.8 | 5.0 | 42.2 | 80 | 1.0 | 2790 | 7000 | 43 | 82 | 125 |
| | 0–20 | 14 | 3.1 | 4.5 | 66.6 | 71 | —— | 2250 | 7250 | 44 | 55.7 | 99.7 |
| NT | 0–5 | 21 | 2.2 | 9.5 | 35.8 | 67 | 1.0 | 1450 | 10,500 | 18 | 34 | 52 |
| | 5–10 | 16 | 1.9 | 8.4 | 75.8 | 80 | 1.1 | 1050 | 8800 | 42 | 44 | 85 |
| | 10–20 | 11 | 1.9 | 6.0 | 36.5 | 72 | 1.2 | 2280 | 13,200 | 43 | 85 | 128 |
| | 0–20 | 15 | 2.0 | 7.5 | 51.2 | 81 | —— | 1765 | 11,425 | 36.5 | 62 | 98.5 |

Sist.: tillage system, OC = organic carbon, TN = total nitrogen, C/N = carbon to nitrogen ratio, BD = bulk density determined by the core method; E$_{NT}$ = total nitrogen stock, E$_{CO}$ = organic carbon stock, E$_{NH4}^+$ = nitrogen stock as ammonium, E$_{NO3}^-$ = nitrogen stock as nitrate, E$_{NM}$ = mineral nitrogen (E$_{NH4+}$ + E$_{NO3}$).

Furthermore, soil samples were collected to determine bulk density, and stocks of total N (Ntotal), total C (Ctotal), and mineral N (Nmineral). Soil bulk density was determined by the core method [28]. Total N was determined by the Kjeldhal method [29]. The mineral N was determined by extraction with 1 mol L$^{-1}$ KCl and magnesium oxides and devarda's alloy [30]. The stocks of total N, mineral N, and total carbon from each soil depth were calculated according to Santos et al. [11] (Table 3).

*2.4. Estimate of N Mineralization in the Soil*

We were interested in monitoring the behavior of N and the fertilizer technologies when applied in both tillage systems. For that, we decided to estimate the mineralization of N in both tillage systems. The objective was to perform a complete characterization of the studied areas and also explain some behaviors of the tillage systems relative to the evaluated agronomic parameters. For this, the estimation of N mineralization was performed as proposed by Brady and Weil [31], with adaptations described in Santos et al. [11].

The data used on this estimate can be found on Table 3. Since both experiments were conducted in soils with clayey texture under tropical conditions, we adopted the value of 3% of annual N mineralization, as proposed by Brady and Weil [31]. The results of the estimate of N mineralization in the studied soils are presented on Table S2.

*2.5. Treatments and Experimental Design*

The experiments consisted of twelve (12) treatments assembled in a $6 \times 2$ factorial scheme (N fertilizers and their technologies applied in the soil as top-dressing fertilization: (1) prilled urea (PU), (2) urea treated with NBPT (N-(n-butyl) thiophosphoric triamide), (3) urea + Cu + B (U$_{CuB}$), (4) ammonium nitrate (AN), (5) ammonium sulfate (AS), and (6) without N application—control; and tillage systems management for corn cultivation: conventional tillage (T) and no-till (NT) (Figure S2). For the two years of experiment, the treatments were composed by three repetitions.

The sowing of corn in the 2017/2018 crop season was performed along with the application of 18 kg N ha$^{-1}$ and 11.4 kg P$_2$O$_5$ ha$^{-1}$ (formula 27-17-00). In the 2018/2019 crop season, the sowing was performed along with 18 kg N ha$^{-1}$ and 32 kg P$_2$O$_5$ ha$^{-1}$ (Bulk blend of fertilizers 14-25-00).

The spacing between rows was 0.75 m, totaling 55,000 plants per hectare. Each experimental plot consisted of six sowing rows with 5 m length each. 150 kg N ha$^{-1}$ were applied via top dressing fertilization. Fertilizers were applied in the sowing lines at a distance of approximately 10 cm from the plant collar. The three central meters and three central lines of each plot (6.75 m$^2$) were considered the useful plot.

2.5.1. Ammonia Volatilization

To quantify the N-NH$_3$ losses, PVC collectors adapted by Lara-Cabezas et al. [32]. As a support of the collectors, three bases of PVC tubes were installed in each experimental

plot at a distance of 10 cm from the corn sowing row. The bases had 12 cm $\times$ 20 cm $\times$ 5 cm (diameter, height, and depth in the soil).

After the application of the treatments in the bases, N-NH$_3$ collectors with dimensions 50 cm $\times$ 12 cm (height and diameter, respectively) were installed. Two sponges (0.02 g cm$^{-3}$ density) soaked with phosphoric acid solution (60 mL L$^{-1}$) and glycerin (50 mL L$^{-1}$) were placed inside each collector.

The sponge located in the upper part of the collector meant to prevent the contamination of the lower sponge with gases from the atmosphere, whereas the sponge at the lower part was used to absorb the ammonia volatilized. In order to reduce the spatial variability of the N-NH$_3$ losses, and to simulate the field conditions, such as temperature and precipitation, the collectors were alternated between the three bases. Thus, after each collection of sponges, the collector was changed from its base.

The N-NH$_3$ collections were carried 1, 2, 3, 4, 5, 7, 10, 14, 19, 23, and 29 days after the application of the treatments in the top-dressing fertilization of corn. The solution in sponges collected in the field was extracted and analyzed as described in Santos et al. [11].

The phosphoric acid solution captures the volatilized ammonia in the form of ammonium phosphate (3 NH$_3$ + H3PO$_4$ $\rightarrow$ (NH$_4$) 3 PO$_4$). The solution present in the sponges collected in the field was extracted by filtration in a Büchner funnel using a vacuum pump, after 10 sequential washes with 40 mL of deionized water. From the extract, aliquots of 20 mL were taken to determine the N concentration by distillation using the Kjeldahl method (Kjeldahl 1883) [29].

After calculating the N levels in the samples, the obtained value (corresponding to the area occupied by the base with the chambers installed in the field) was extrapolated to the percentage of N-NH$_3$ loss per hectare. To calculate the accumulated losses during the 29 days, losses from the 1st and the 2nd day were added; the sum of these values was then added to the 3rd day, and so on. During the period of evaluation of N-NH$_3$ losses, the climate data were collected by the automatic weather station from the Brazilian Ministry of Agriculture, Livestock, and Supply (MAPA), located in Bambuí, Minas Gerais State, Brazil.

### 2.5.2. Weather Conditions

Data on rainfall, relative air humidity, and maximum and minimum temperature were recorded by the meteorological station of the farm. Data were collected throughout the entire period of evaluation of N-NH$_3$ losses by volatilization. Rainfall, maximum and minimum temperature values, and relative air humidity after 29 days of the application of top-dressing N fertilization in both experiments in Medeiros e Bambuí in the 2017/2018 and 2018/2019 crop season are presented on Figure S3.

In the 2017/2018 crop season, precipitation values of 45, 18, 10, and 42.5 mm occurred in the first seven days after the application of the N fertilizers, totaling 115.5 mm of precipitation; the mean temperature was 23.5 °C. As for the 2018/2019 crop season, precipitation values of 7.5, 24, 11, 41, and 5 mm occurred in the first seven days after the application of the treatments, totaling 80.5 mm; the mean temperature was 23 °C. During the entire growth cycle of the corn, the precipitation was 435 mm and 372 in the 2017/2018 and 2018/2019 crop seasons, respectively.

### 2.5.3. Nitrogen Accumulation and Corn Yield

When the corn grains reached the physiological maturity, the corn cobs were harvested and separated from the culm and leaves (which correspond to the straw). The grains were removed from the cobs using a thresher and then, the grain moisture was quantified using a Gehaka® equipment G600 for subsequent correction of moisture to 13%. Then, this value was extrapolated to represent the grain yield in kg ha$^{-1}$. From this sample of grains, a subsample was taken and oven dried at 65 °C for subsequent analysis of N content in the grains.

To estimate the straw production, the samples were weighed, grinded in a forage harvester, and then, subsamples were taken for determination of moisture content. Afterwards, the results were extrapolated to calculate the production of straw per hectare, and the values were given in kg ha$^{-1}$. Similar to the grains, subsamples of straw were dried and grinded in a Willey mill for analysis of N content by the Kjeldahl method [29] and following the methodology described by Tedesco et al. [30].

*2.6. Statistical Analysis*

The treatments were submitted to a non-linear regression analysis using a logistic model to evaluate the losses of ammonia by volatilization, Equation (1):

$$Yi = \left[ \frac{\alpha}{1 + e^k (b - daai)} \right] + Ei \tag{1}$$

In which, *Yi* is the i-th observation of the accumulated loss of N-NH$_3$ in %, being *i* = 1, 2, . . . , *n*; *daa$_i$* is the i-th day after the application of the treatment; $\alpha$ is the asymptotic value that can be interpreted as the maximum amount of accumulated loss of N-NH$_3$; *b* is the abscissa of the inflection point and indicates the day when the maximum loss by volatilization occurs; *k* is the value that represents the precocity index, and the higher its value, the lower the time needed to reach the maximum loss by volatilization ($\alpha$); *E$_i$* is the error associated to the i-th observation, which is assumed to be independent and equally distributed according to a zero average standard and constant variance, $E \sim N (0, I \sigma^2)$.

This model has been largely applied to estimate plant growth, and recently, has been used to estimate the accumulated loss of N-NH$_3$ [19,33,34].

To estimate the maximum daily loss (day when the highest loss of N-NH$_3$ occurred), that is, to determine the inflection point of the curve, it was used the following Equation (2):

$$PMD = k \times (\alpha/4) \tag{2}$$

In which, *k* is a relative index used to obtain to maximum daily loss (MDL), and $\alpha$ is the asymptotic value that can be interpreted as the maximum amount of accumulated loss of N-NH$_3$.

Analysis of variance was applied to test the influence of the fertilizers in the following parameters: accumulated losses of ammonia by volatilization at the end of the evaluation days, grain yield, straw production, and N removal. The significance of the differences was evaluated in $p \leq 0.05$. After validating the statistic model, the mean values were grouped by the Scott–Knott algorithm using the R software 3.3.1 [35].

## 3. Results

*3.1. Ammonia Volatilization*

The daily losses of N-NH$_3$ in the 2017/2018 crop season varied according to the N fertilizers applied in the NT system compared to the conventional tillage (Figure 2).

In the conventional system, the maximum loss of N-NH$_3$ was 21 kg ha$^{-1}$ at 2.4 days for PU (Table 4, Figure 2). For the U$_{CuB}$ and U$_{NBPT}$ (180 mg kg$^{-1}$) treatments, the maximum losses were 6.8 and 1.6 kg N at 3.1 and 5.7 days after fertilization, respectively. The efficiency of these treatments to reduce N-NH$_3$ losses is evidenced by the time in days for these treatments to reach 10, 20, and 50% of the maximum losses for PU, which were 1.6, 2.4, and 4.2 days for U$_{CuB}$ and 2.2, 5.1 for U$_{NBPT}$. For 50% of the losses, U$_{NBPT}$ did not reach 50% of the maximum loss of PU (Table 4).

In the NT system, the maximum loss for PU was 13.6 kg ha$^{-1}$ at 2.3 days after fertilization. The maximum loss values for the U$_{CuB}$ and U$_{NBPT}$ (180 mg kg$^{-1}$) treatments were 9.4 and 7.4 kg ha$^{-1}$ at 2.6 and 3.3 days after fertilization, respectively (Table 4, Figure 2). In this case, the time in days for the occurrence of 10, 20, and 50% of the maximum losses for urea were 0.1, 1.1, and 2.6 days for U$_{CuB}$ and 0.9, 2, and 4 days for U$_{NBPT}$ (Table 4). The

AS and AN treatments had maximum losses between 2 and 7 days in both systems, but with values lower than 0.5 kg N ha$^{-1}$.

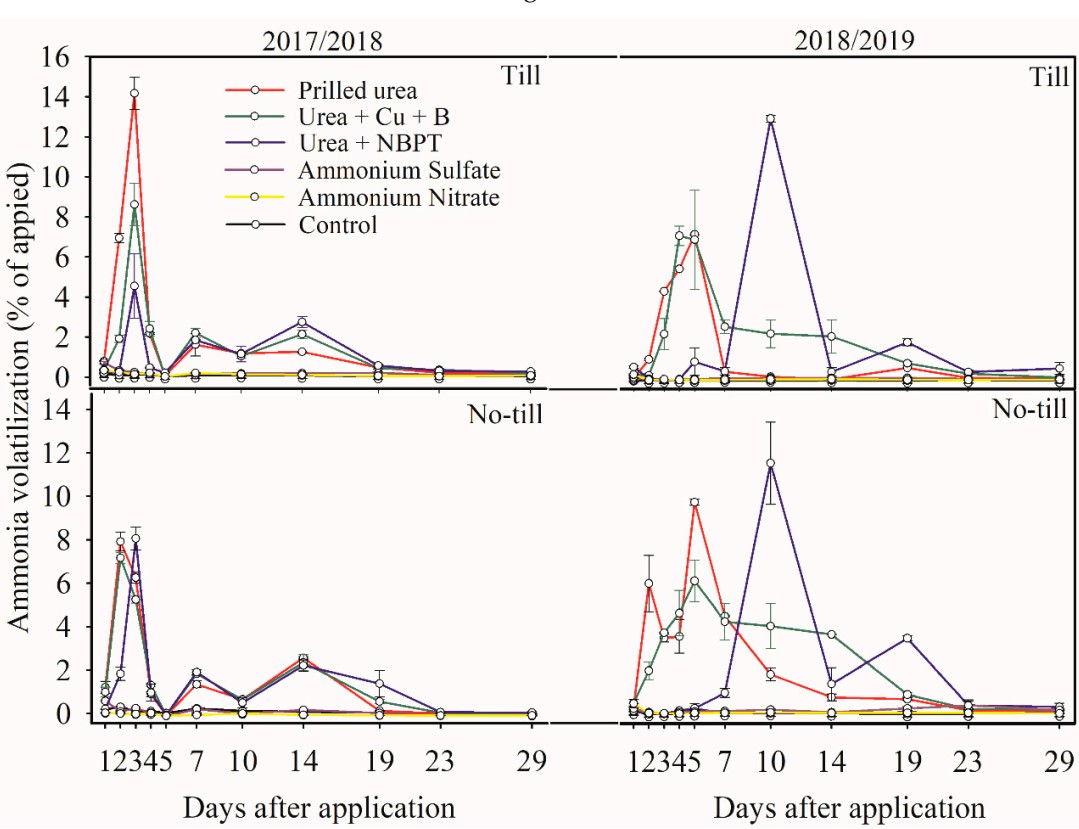

**Figure 2.** Daily losses of N-NH$_3$ of the fertilizer sources applied as topdressing during corn cultivation in the 2017/2018 and 2018/2019 crop seasons, in conventional tillage (T) and no-till (NT) systems.

The percentages of N-NH$_3$ losses that occurred in the first seven days in the conventional tillage system were 89, 79, and 60% for the PU, U$_{CuB}$ and U$_{NBPT}$ treatments, respectively. As for NT, these values were 82, 80, and 74%, for the PU, U$_{CuB}$ and U$_{NBPT}$ treatments, respectively. Thus, more than 80% of the N-NH$_3$ losses occurred during the first seven days after the application of PU (Table 4).

In the 2018/2019 crop season, the N-NH$_3$ losses were affected by the applied N sources and cropping systems (Table 4, Figure 2). In the conventional system, the maximum N-NH$_3$ loss was 11.4 kg ha$^{-1}$ for PU at 3.6 days. The U$_{CuB}$ and U$_{NBPT}$ treatments had losses of 10.6 and 8 kg N ha$^{-1}$ at 4.3 and 8.6 days after application, respectively. The time needed to reach 10, 20, and 50% of the maximum losses for PU were 2.3, 3, and 4 days for U$_{CuB}$ and 7, 7.6, and 9 days for U$_{NBPT}$. Showing that the U$_{NBPT}$ has greater efficiency in reducing losses than the U$_{CuB}$. As for the losses in the NT system, the day of maximum PU loss was similar to the conventional system (9.6 kg ha$^{-1}$ at 3.9 days). In the U$_{CuB}$ and U$_{NBPT}$ treatments, the maximum losses occurred at 5 and 9 days, with values of 6.6 and 6.3 kg of N ha$^{-1}$, respectively. However, the time needed to reach 10, 20, and 50% of the maximum loss for PU was higher than the conventional system, with values of 1.7, 2.7, and 5 days for U$_{CuB}$ and 7.5, 8.3, and 11 days for U$_{NBPT}$ (Table 4). As for the 2017/2018 crop season, the AS and AN treatments had maximum losses between 1 and 14 days, but with values below 0.2 kg of N ha$^{-1}$ (Table 4, Figure 2).

In the conventional tillage system, PU had the higher percentage of volatilized N-NH$_3$ in the first seven days (92%), followed by U$_{CuB}$ and U$_{NBPT}$, with 67% and 8%, respectively. As for the NT system, the accumulated losses in the first seven days were 88% (PU), 70% (U$_{CuB}$) and 10% (U$_{NBPT}$). Similar to the 2017/2018 crop season, the losses for PU reached approximately 90% until the 7th day relatively to the 29 days of collection.

**Table 4.** Regression parameters adjusted to the accumulated losses of N-NH$_3$ by volatilization and maximum daily losses of the fertilizers under conventional and NT systems.

| Treatment | System | Parameters | | | | MDL (kg) | Time for Volatilization of 10, 20 and 50% of the Maximum Losses Observed for PU | | |
|---|---|---|---|---|---|---|---|---|---|
| | | $\alpha$ | b | k | $R^2$ | | 10% (Day) | 20% (Day) | 50% (Day) |
| | | | | | Crop season 2017/2018 | | | | |
| PU | NT | 30.27 | 2.30 | 1.20 | 0.91 | 13.62 | 0.5 | 1.2 | 2.3 |
| | T | 27.17 | 2.41 | 2.10 | 0.97 | 21.39 | 1.3 | 1.7 | 2.4 |
| U$_{CuB}$ | NT | 29.34 | 2.59 | 0.86 | 0.91 | 9.46 | 0.1 | 1.1 | 2.6 |
| | T | 18.16 | 3.17 | 1.01 | 0.93 | 6.87 | 1.6 | 2.4 | 4.2 |
| U$_{NBPT}$ | NT | 24.78 | 3.31 | 0.80 | 0.89 | 7.43 | 0.9 | 2.0 | 4.0 |
| | T | 12.13 | 5.71 | 0.36 | 0.93 | 1.63 | 2.2 | 5.1 | ** |
| AS | NT | 3.64 | 3.33 | 0.26 | 0.97 | 0.35 | - | - | - |
| | T | 2.33 | 2.90 | 0.19 | 0.96 | 0.16 | - | - | - |
| AN | NT | 1.38 | 3.97 | 0.33 | 0.96 | 0.17 | - | - | - |
| | T | 1.49 | 3.34 | 0.36 | 0.97 | 0.20 | - | - | - |
| Control | NT | 1.26 | 3.61 | 0.38 | 0.97 | 0.17 | - | - | - |
| | T | 0.84 | 4.03 | 0.35 | 0.97 | 0.11 | - | - | - |
| | | | | | Crop season 2018/2019 | | | | |
| PU | NT | 30.50 | 3.98 | 0.84 | 0.98 | 9.60 | 1.5 | 2,3 | 4.0 |
| | T | 19.52 | 3.69 | 1.56 | 0.99 | 11.41 | 2.3 | 2.8 | 3.7 |
| U$_{CuB}$ | NT | 29.08 | 4.98 | 0.61 | 0.98 | 6.65 | 1.7 | 2.7 | 5.0 |
| | T | 23.98 | 4.37 | 1.18 | 0.97 | 10.61 | 2.3 | 3.0 | 4.0 |
| U$_{NBPT}$ | NT | 17.70 | 8.94 | 0.95 | 0.98 | 6.30 | 7.5 | 8.3 | 11 |
| | T | 17.02 | 8.57 | 1.26 | 0.99 | 8.04 | 7.0 | 7.6 | 9.0 |
| AS | NT | 0.92 | 4.78 | 0.36 | 0.94 | 0.12 | - | - | - |
| | T | 1.50 | 1.09 | 0.13 | 0.97 | 0.07 | - | - | - |
| AN | NT | 1.18 | 2.27 | 0.15 | 0.99 | 0.06 | - | - | - |
| | T | 0.89 | 3.90 | 0.16 | 0.98 | 0.05 | - | - | - |
| Control | NT | 0.62 | 5.07 | 0.25 | 0.98 | 0.05 | - | - | - |
| | T | 0.30 | 6.43 | 0.22 | 0.90 | 0.02 | - | - | - |

$\alpha$: Asymptotic value (percentage of maximum volatilization); b: Day when the maximum N-NH$_3$ loss occurs; k: relative index and MDL (maximum daily loss); ** The maximum loss of this treatment did not reach 50% of the maximum loss for PU.

The accumulated N-NH$_3$ losses were affected ($p < 0.05$) by the interaction between the applied fertilizers and the cropping systems in the 2017/2018 crop season. Regardless of the fertilizer and the cultivation system, there is a higher loss of N-NH$_3$ in the first days after fertilizer application; however, for U$_{NBPT}$ and U$_{CuB}$, these values are low until approximately 5 to 9 days after fertilizer application (Figure 3). On the other hand, the behavior of the AN and AS treatments had the same tendency to control, without the application of N (Figure 3). The mean accumulated losses of N-NH$_3$ by volatilization under NT (21.6 kg N ha$^{-1}$) were 49% greater than in the conventional tillage (14.5 kg N ha$^{-1}$) (Figure 3).

In the 2017/2018 season, the accumulated N-NH$_3$ losses presented the following decreasing order for the NT system: PU (33 kg ha$^{-1}$) = U$_{CuB}$ (31 kg ha$^{-1}$) > U$_{NBPT}$ (27 kg ha$^{-1}$) > AS (3.7 kg ha$^{-1}$) = NA (1.5 kg ha$^{-1}$). In the conventional system, the accumulated losses decreased as follows: PU (29 kg ha$^{-1}$) > U$_{CuB}$ (19.7 kg ha$^{-1}$) > U$_{NBPT}$ (12.6 kg ha$^{-1}$) > AS (2.4 kg ha$^{-1}$) = AN (1.6 kg ha$^{-1}$) (Table 5, Figure 3). When considering the influence of the cropping systems on the accumulated N-NH$_3$ losses, only the U$_{NBPT}$ and U$_{CuB}$ treatments showed higher losses in the NT system (53 and 37%, respectively) compared to the conventional planting system.

In the 2018/2019 season, accumulated N-NH$_3$ losses were also affected ($p < 0.05$) by the interaction between the applied fertilizers and the cropping systems. The mean accumulated N-NH$_3$ losses under NT were 26.2% greater than the losses observed under conventional tillage (Figure 3).

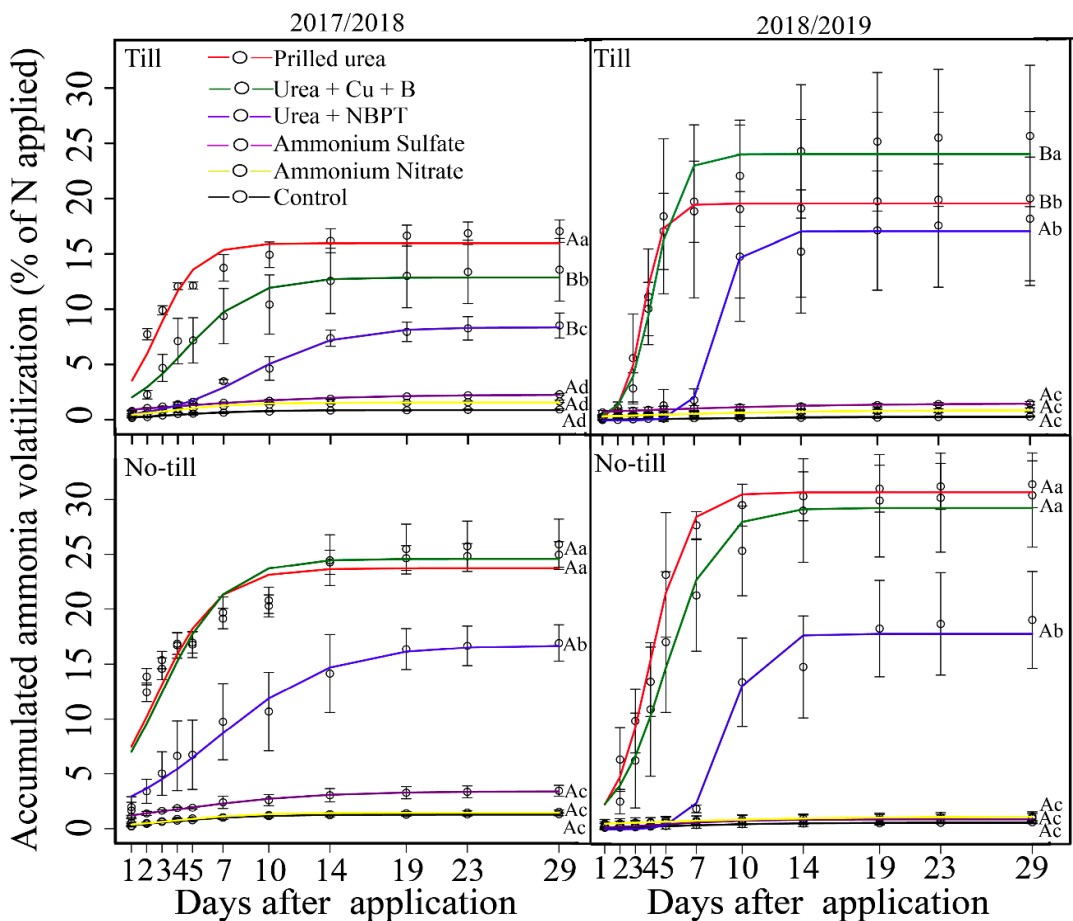

**Figure 3.** Accumulated losses of N-NH$_3$ by volatilization per fertilizers applied as top-dressing fertilization of corn in the 2017/2018 and 2018/2019 crop seasons, under conventional tillage (T) and no-till (NT) systems. Means followed by the same upper letter (tillage system) and lower letter (sources of N fertilizer) do not differ at 5% significance level by the Scott–Knott test.

**Table 5.** Accumulated losses under NT and conventional tillage systems, accumulated losses, and percentage increase in the losses under NT relatively to conventional tillage.

| | No-till | | | | | |
|---|---|---|---|---|---|---|
| **Treatment** | ── 2017/2018 ── | | ── 2018/2019 ── | | ─── Sum ─── | | |
| | **NH$_3$ (kg ha$^{-1}$)** | **% Red.** | **NH$_3$ (kg ha$^{-1}$)** | **% Red.** | **NH$_3$ (kg ha$^{-1}$)** | **% Red.** | **% >PD/PC (%)** |
| U$_{NBPT}$ | 27.4 Ab | −17 | 18.9 Ab | −39 | 46.3 Ab | −28 | +34 |
| U$_{CuB}$ | 31.7 Aa | −4 | 30.2 Aa | −3 | 61.9 Aa | −3 | +27 |
| AN | 1.5 Ac | −95 | 1.0 Ac | −96 | 2.7 Ac | −96 | +7 |
| AS | 3.7 Ac | −89 | 1.2 Ac | −97 | 4.8 Ac | −92 | +17 |
| PU | 32.9 Aa | ── | 31.2 Aa | ── | 64.1 Aa | ── | +23 |
| CV(%) | 11 | ── | 18 | ── | 13 | ── | ── |
| | | | Till | | | | |
| U$_{NBPT}$ | 12.6 Bc | −57 | 18.5 Ab | −6 | 30.7 Bb | −37 | ── |
| U$_{CuB}$ | 19.8 Bb | −32 | 25.6 Ba | +28 | 45.3 Ba | −7 | ── |
| AN | 1.6 Ad | −94 | 0.9 Ac | −95 | 2.5 Ac | −95 | ── |
| AS | 2.4 Ad | −92 | 1.5 Ac | −92 | 4.0 Ac | −92 | ── |
| PU | 29.0 Aa | ── | 20.0 Bb | ── | 49.0 Ba | ── | ── |
| CV(%) | 11 | ── | 18 | ── | 13 | ── | ── |

NH$_3$ = losses of N-NH$_3$ by volatilization, % Red = percentage of reduction in the N-NH$_3$ losses in relation to PU, >PD/PC = percentage of increase in the N-NH$_3$ losses under NT relatively to conventional tillage. Means followed by the same upper letter do not differ between tillage systems, and lower letters do not differ between the studied sources.

The accumulated N-NH$_3$ losses for NT decreased as follows: PU (30.2 kg ha$^{-1}$) = U$_{CuB}$ (31.2 kg ha$^{-1}$) > U$_{NBPT}$ (19 kg ha$^{-1}$) > AN (1.0 kg ha$^{-1}$) − 1) = AS (1.2 kg ha$^{-1}$). As for the conventional system, the accumulated N-NH$_3$ losses decreased in the following order: U$_{CuB}$ (25.6 kg ha$^{-1}$) > PU (20 kg ha$^{-1}$) = U$_{NBPT}$ (18 kg ha$^{-1}$) > AS (1.5 kg ha$^{-1}$) = AN (0.9 kg ha$^{-1}$) (Table 5, Figure 3). The accumulated N-NH$_3$ losses were 15 and 36% higher in the NT system than in the conventional system for the U$_{CuB}$ and PU treatments, whereas the other treatments did not differ. In both crop seasons and cropping systems, the accumulated N-NH$_3$ losses in the AS and NA treatments were equal to the control treatment, without N application.

For a better understanding of the studied technologies within cropping systems, the losses from both crop seasons were added, and using these values, we calculated the percentage increase of the losses under NT relatively to conventional tillage (Table 5). The percentage increase in the losses of NT > T varied between 7 and 34% (Table 5).

### 3.2. Effects of Fertilizers and Soil Nitrogen Stocks on Nutrient Accumulation and Corn Yield

For the 2017/2018 crop season, the N extraction by the grains (Figure S4), corn straw (straw), and total extraction (grains + straw) were not influenced ($p \geq 0.05$) by the interaction between N fertilizer and tillage systems. When evaluating these factors separately, there was an effect ($p \leq 0.05$) on the N extraction by the corn straw (straw) and total extraction (grains + straw) as a function of the N sources. Difference in N extraction values were observed only between the N sources applied relative to the control, without N application (Figure S4).

The extraction of N by corn grains in the 2018/2019 crop season was not affected ($p \geq 0.05$) by the interaction between sources and cropping systems, nor by the isolated effect of these factors. The mean N extraction by the grains was 182 kg of N ha$^{-1}$ (Figure S5A). The N extraction by straw was not influenced by the interaction between fertilizers sources and tillage systems ($p \geq 0.05$); only by the effect of fertilizer sources (Figure S5A). The lowest N extraction by the straw occurred in the control treatment (44 kg ha$^{-1}$) (Figure S5A). The total N extraction (grains + straw) was influenced by the interaction between tillage systems and N sources ($p \leq 0.05$). The lowest total N extraction was observed in the control (199 kg ha$^{-1}$) and U$_{CuB}$ (224 kg ha$^{-1}$) treatments under NT; the other treatments did not differ (Figure S5B). Regarding the tillage systems, there was a difference only for U$_{CuB}$, with increased N extraction (318 kg ha$^{-1}$) under NT (Figure S5B).

Corn grain yield and production of straw were not affected by the N sources applied and the tillage system (T and NT) in both crop seasons ($p \geq 0.05$) (Figure S6).

In the 2017/2018 crop season, the average grain yield of the N sources varied between 9532 and 10,982 kg ha$^{-1}$ under conventional tillage, and between 8914 and 10,895 kg ha$^{-1}$ under NT. The straw production varied between 6431 e 7513 kg ha$^{-1}$ under conventional tillage, and between 6124 and 6988 kg ha$^{-1}$ under NT (Figure S6B).

In the 2018/2019 crop season, the average values observed in the studied N sources ranged between 11,622 and 15,795 kg ha$^{-1}$ under conventional tillage, and between 11,533 and 15,799 kg ha$^{-1}$ under NT (Figure S6C). The average straw production in the 2018/2019 crop season ranged between 8634 and 11,600 kg ha$^{-1}$ under conventional tillage, and between 7848 and 10,948 kg ha$^{-1}$ under NT (Figure S6D).

Although it is not possible to statistically analyze the data, in Table 6, we have the balances between the inputs and outputs of N of the system, as well as the remaining N for the subsequent cultivation. Another important aspect to mention is that, prior to the cultivation of corn, the areas were cultivated with soybeans, and leaves for the successor cultivation about 17 kg of N per ton of grains produced.

Based on the data in Table 6, it is possible to notice that the total N input for corn was between 205 and 271 kg N ha$^{-1}$. In general, for the two crops and cultivation system, the highest N losses of the system were for PU, on average 24 kg ha$^{-1}$, for conventional and 32 kg ha$^{-1}$ for no-tillage.

**Table 6.** Systems nitrogen balance.

| System | | TN | Min. N | E-NO$_3^-$ | E-NH$_4^+$ | N Fertlization | Full Availability | N-NH$_3$ Loss | Grain Extraction | N Residual Soil |
|---|---|---|---|---|---|---|---|---|---|---|
| | | | **Soil N Availability** | | | | | | | |
| **2017–2018** | U$_{NBPT}$ | | | | | | | 12.6 | 142.3 | 21.6 |
| | U$_{CuB}$ | | | | | | | 19.8 | 147 | 38.2 |
| | AN | 2330 | 22.5 | 24.5 | 8 | 150 | 205 | 1.6 | 143 | 60.4 |
| | AS | | | | | | | 2.4 | 164.6 | 38 |
| | PU | | | | | | | 29 | 145 | 31 |
| | U$_{NBPT}$ | | | | | | | 27.4 | 155.5 | 25.8 |
| | U$_{CuB}$ | | | | | | | 31.7 | 144.2 | 32.8 |
| | AN | 2024 | 20 | 11 | 27.7 | 150 | 208.7 | 1.5 | 137.8 | 69.4 |
| | AS | | | | | | | 3.7 | 146.5 | 58.5 |
| | PU | | | | | | | 32.9 | 164.5 | 11.3 |
| **2018–2019** | U$_{NBPT}$ | | | | | | | 18.5 | 174.3 | 78.6 |
| | U$_{CuB}$ | | | | | | | 25.6 | 225 | 20.8 |
| | AN | 2250 | 21.7 | 55.7 | 44 | 150 | 271.4 | 0.9 | 173.5 | 97 |
| | AS | | | | | | | 1.5 | 186.7 | 83.2 |
| | PU | | | | | | | 20 | 179.4 | 72 |
| | U$_{NBPT}$ | | | | | | | 18.9 | 228.5 | 17.6 |
| | U$_{CuB}$ | | | | | | | 30.2 | 157 | 77.8 |
| | AN | 1765 | 16.5 | 62 | 36.5 | 150 | 265 | 1 | 186.5 | 77.5 |
| | AS | | | | | | | 1.2 | 201 | 62.8 |
| | PU | | | | | | | 31.2 | 189 | 44.8 |

TN = total nitrogen, Min. N = Potentially available nitrogen, since it will depend on the mineralization rate, E-NH$_4^+$ = nitrogen stock as ammonium, ENO$_3^-$ = nitrogen stock as nitrate.

The residual N in the system was higher in both crops and systems, for NA (mean of 78.7 kg for conventional and 73.4 kg for no-tillage). The smallest residual was left by the U$_{CuB}$, with an average value of 29.5 kg ha$^{-1}$ for conventional tillage. The residual N values for U$_{NBPT}$, although it presented the lowest losses of N by volatilization between treatments, was on average 64.3 kg ha$^{-1}$ for conventional and 21.7 kg ha$^{-1}$ for no-tillage.

## 4. Discussion

Observing the behavior of the evaluated N fertilizers regarding the daily losses, we noted that the highest values in both crop seasons occurred with the application of PU, approximately 2.5 days after the application in both crop seasons and cropping systems. This occurs because, when applied to the soil, without any additive or technology that reduces its solubility or the hydrolysis rate, urea is easily degraded by urease into NH$_3$ and CO$_2$ [10,14]. If not incorporated, this ammonia becomes susceptible to losses by volatilization.

In our study, rainfall up to the second day after fertilization was 85 and 28 mm in the first and second year, respectively. However, it is complicated to accurately inform the amount and intensity of rainfall needed to incorporate urea into the soil since the values obtained were insufficient. Similar to what was observed in the application of PU, several studies also demonstrate that the maximum daily loss of N-NH$_3$ occurs in the first days following the application of the fertilizers [11,36–38]. Thus, we can argue that these losses will occur in the first days for urea without any treatment or technology, as long as the moisture conditions allow the hydrolysis process. The moisture conditions do not rely only on rainfall since a value of relative air humidity above 74.3% (critical humidity of urea at 30 °C) can already start the hydrolysis process [4]. In our study, in the first crop season, the mean temperature was higher than 30 °C, and the relative air humidity was higher than the critical humidity of urea in both crop seasons. Such increased air humidity can promote increased N-NH$_3$ losses, even without rainfall.

The delay in the day of maximum loss observed in both systems for the stabilized fertilizers ($U_{CuB}$ and $U_{NBPT}$) compared to PU is due to the inhibition mechanism of each technology. For $U_{NBPT}$, this reduction in urease activity is due to the ability of NBPT to be oxidized into its analog compound NBPTO, which can inhibit the urease activity by forming stable complexes with the enzyme [39]. As for $U_{CuB}$, the urease activity is inhibited due to the binding of Cu to the sulfhydryl group. Such binding blocks the active site of the enzyme, preventing the urea molecule to bind to the sulfhydryl group. Thus, the process of urea hydrolysis cannot occur [40]. The effect of B on urease inhibition diverges among different authors, but according to Santos et al. [11], the study by Benini et al. [39] provides a better explanation. These authors attribute the efficiency of B to its competitive inhibition when binding between the Ni ions of the enzyme, where the urea molecule would bind, which prevents the hydrolysis process from occurring.

The delayed urea hydrolysis when using these two technologies may favor the incorporation of fertilizers into the soil after subsequent precipitation events, which may reduce N-NH$_3$ losses. This effect was observed for both sources in the 2017/2018 crop season. After the fifth day, which had a 42 mm rainfall, fertilizers were probably incorporated into the soil. Then, the daily N-NH$_3$ losses decreased from that point.

The inhibition of urease by NBPT, indirectly observed by the N-NH$_3$ losses, occurs in varying intensities between the NT and conventional systems. In our study, such behavior is evidenced by increasing the NBPT concentration in urea from 180 mg kg$^{-1}$ to 1200 mg kg$^{-1}$ (2017/2018) in the second year of the experiment. We noticed that the day of the maximum loss for the $U_{NBPT}$ treatment (1200 mg kg$^{-1}$) under conventional tillage was delayed, occurring at the 8.5th day after application. Although it is not possible to compare the two seasons under study, this result represents a delay of 67% relatively to the previous crop season. When observing the day when the maximum loss occurred under NT, no differences occurred between the tillage systems, thus we can infer that this concentration was efficient for both cropping systems.

However, we cannot interpret this concentration (1200 mg kg$^{-1}$) as adequate for the treatment of urea to be used in both systems. Theoretically, this concentration can be lower in the conventional tillage, which also reflects the efficient use of the inhibitor. These results show that the amount of NBPT used in the treatment of urea may need an adjustment as a function of the soil and cropping conditions, that is, more precise information about the relationship between the NBPT concentration and the values of urease activity is needed.

Despite the positive results reported on the use of Cu and B, in the 2017/2018 crop season, we observed that the maximum loss of $U_{CuB}$ occurred in a time frame (days) similar to PU under conventional tillage. The lower efficiency of $U_{CuB}$ (only 3% relatively to PU) to reduce N-NH$_3$ losses can be explained by the low concentration of micronutrients (Cu and B) in the fertilizer. Furthermore, it should be emphasized that the amount of metallic cations and compounds containing mostly B added to urea aiming the inhibition of urease activity should be carefully evaluated. In this study, 0.3% of Cu (copper sulfate) and 0.3% of B (boric acid) were added to urea, which, considering the dose of 150 kg ha$^{-1}$, correspond to 450 g of Cu and B in the region of the dissolution of urea.

The other sources used in this study (AS, AN) did not promote significant daily losses. Such reduced losses are due to the N form present in the AS and AN fertilizers, and also due to their acidic reaction, which creates a less favorable environment to the N-NH$_3$ losses by volatilization, as previously reported in several studies [4,41,42].

The accumulated N-NH$_3$ losses were higher under NT than under conventional tillage in both crop seasons. This occurred as a result of the greater presence of crop residues (straw) in that system, which favors the rapid hydrolysis of the fertilizer due to the increased urease activity [43,44]. Moreover, the crop residues present in the NT system reduce the diffusion of urea in the soil by reducing the contact urea/soil [2,7,15].

Our findings demonstrate alternatives to reduce urease activity and $N-NH_3$ losses in NT systems, which would be the use of technologies that enhance urea use efficiency. Thus, $U_{NBPT}$ stands out as the best technology as it reduced, on average, 28% of losses relatively to PU in the NT system for both crop seasons. Another technology that may be used is the coating with metallic ions and compounds containing B. However, the Cu and B concentrations deserve further investigation since, in our study, the reduction in $N-NH_3$ losses was only 3% on average for both crop seasons.

Owing to their acidic reaction, the AS and AN sources presented the lowest accumulated losses. The accumulated losses observed for these sources (lower than 0.5%) are already reported in several studies conducted in soils cultivated under this pH range [2,4,44]. These values of $N-NH_3$ losses quantified for AS and AN do not actually represent the losses from these fertilizers, since they are equal to the values observed in the soil without N application. This indicates that these losses occur naturally in the soils, even in the control treatment without N fertilization.

The reductions in $N-NH_3$ losses by volatilization were not followed by expressive increases in N extraction by corn. In both crop seasons, the low extraction by corn straw in the control treatment is due to the absence of N fertilization.

The grain yield in both crop seasons was twice the average Brazilian yield (5029 kg ha$^{-1}$) [45]. The absence of responses regarding treatments and cropping systems is related to the high N supply by the soil. This is due to the increased N stock in the soil (67 kg ha$^{-1}$ on average), and also to the potential of N mineralization in the soil (60 kg ha$^{-1}$ year$^{-1}$ on average), which has been under NT for at least 15 years.

Other results on grain yield, N extraction by the grains and straw show that there is reduction in ammonia losses with the use of inhibitors and other technologies for fertilizers. However, these losses were not followed by increased N extraction by the grains, straw and yield [14,46]. In addition, such studies did not estimate the potential supply of N by the soil organic matter.

In order to explain this lack of response to the N fertilization, interpreting the data on Table S2, we can observe that this soil had the potential supply of approximately 100 kg N ha$^{-1}$ in the 2017/2018 crop season and approximately 156 kg N ha$^{-1}$ in 2018/2019. Thus, the application of 150 kg N ha$^{-1}$ would hardly present a response in yield. These results show that, from an economic viewpoint, the reduction in losses has little effect on crop productivity. However, the maintenance of N in the soil organic matter is as important as the increase in crop productivity. Besides representing a future N supply for crops, the maintenance of N in soil organic matter can mitigate the emission of greenhouse gases into the atmosphere.

In this sense, when we observe the balance between the inputs and outputs of N, the highest residual for AN in both crop season and cultivation is due to the the that this source presents low losses of N by volatilization, since it does not depend on the climatic conditions to define its efficiency in these losses.

Another interesting aspect when we make this balance is that the fact that the PU has left an average residual is mainly due to the greater contribution of N by the organic matter of the soil in the crop season 2018–2019. In addition, although for $U_{CuB}$, in conventional planting it left little residual N in the soil, in no-tillage it was one of the highest values, so, these results confirm that this treatment needs adjustments to increase its efficiency in the system. In addition, with $U_{NBPT}$, the higher extraction of N by the grains in both crops, in a no-tillage system, contributed to the residual N values being the lowest among the treatments. However, these values were a reflection of the small increase in productivity for this source (but without statistical difference).

Another aspect to be taken into consideration is that in our study, the losses of N by $N_2O$ and $NO_3$ were not quantified; however, these values are around 1% [47] and 9% [48] of the applied N, respectively. In this sense, we have mean values of losses of the applied N, around 1.5 and 13.5 for $N_2O$ and $NO_3^{-}$, respectively.

## 5. Conclusions and Future Perspectives

The technologies for urea reduce the N-NH$_3$ losses compared to PU in both studied systems, and the losses under NT are higher than in conventional system. Urea treated with NBPT (1200 mg kg$^{-1}$) is an alternative technology for the efficient N use in grain production systems under NT, as it causes a 5-day delay in the day of maximum loss compared to urea treated with NBPT (180 mg kg$^{-1}$). Ammonium nitrate and sulfate also represent adequate choices to reduce the N-NH$_3$ losses in grain production systems. In the present study, a reduction in N-NH$_3$ losses does not directly reflect an increase in yield and N extraction by corn.

Based on the results observed in this study, we noticed that the NBPT concentration to be used in soils under NT should be adjusted. Thus, studies that evaluate increasing NBPT concentrations in NT systems will be performed by our research group in order to better define the NBPT dose in formulations according to the varying conditions of grain production in tropical regions.

**Supplementary Materials:** The following are available online at https://www.mdpi.com/article/10.3390/soilsystems7020054/s1, Figure S1: Scheme of the simulation of conventional tillage within a no-till area before the sowing of soybean. The authors declare that they own this image., Figure S2: Design of the distribution of treatments in field, referring to only one block of the experiment. NT: no-till system, T: conventional tillage system, T1: control, T2: PU, T3: U$_{NBPT}$, T4: U$_{CuB}$, T5: AN, and T6: AS., Figure S3: Rainfall, maximum and minimum temperatures, and relative air humidity during the evaluation period of the losses of N-NH$_3$ by volatilization in the 2017/2018 (a) and 2018/2019 (b) crop seasons., Figure S4: Nitrogen extraction by the corn grains, shoot dry matter (straw), and total dry matter of corn that received N fertilization in the 2017/2018 crop season. *Treatments followed by the same letter do not differ at 5 % significance level by the Scott-Knott test. The vertical bars indicate the standard error of the mean ($n = 3$)., Figure S5: Nitrogen extraction by the corn grains, shoot dry matter (straw) (a), and total dry matter of corn that received N fertilization (b) in the 2018/2019 crop season. *Treatments followed by the same letter do not differ at 5 % significance level by the Scott-Knott test. The vertical bars indicate the standard error of the mean ($n = 3$)., Figure S6: Corn grain yield and straw production in the 2017/2018 (a and b) and 2018/2019 (c and d) crop seasons that received N top dressing fertilization. Treatments followed by the same upper letter in the bars do not differ within tillage systems (NT and T) and followed by the same lower letter do not differ within N sources at 5 % significance level by the Scott-Knott test. The vertical bars indicate the standard error of the mean ($n = 3$)., Table S1: Soil chemical attributes before the installation of the experiment with 2017/2018 (1°) and 2018/2019 (2°) crop seasons., Table S2: Estimate of the annual mineralization and total availability of N in the studied areas.

**Author Contributions:** Conceptualization, D.G., C.S. and S.I.d.C.P.; methodology, D.G., C.S. and S.I.d.C.P.; software, T.J.F., C.S.; validation, D.G., S.I.d.C.P., A.P.P.N., A.B.d.F. and K.P.e.S.; formal analysis, T.J.F.; investigation, C.S., L.B.S. and R.A.F.; resources, S.I.d.C.P. and D.G.; data curation, D.G., C.S. and S.I.d.C.P.; writing—original draft preparation, C.S., S.D.R., A.B.d.F. and D.G.; writing—review and editing, D.G., S.D.R., C.S. and A.B.d.F.; visualization, D.G., C.S., K.P.e.S and S.I.d.C.P.; supervision, D.G. and S.I.d.C.P.; project administration, S.I.d.C.P. and D.G.; funding acquisition, D.G. and S.I.d.C.P. All authors have read and agreed to the published version of the manuscript.

**Funding:** This research received no external funding.

**Institutional Review Board Statement:** Not applicable.

**Informed Consent Statement:** Not applicable.

**Data Availability Statement:** Not applicable.

**Acknowledgments:** The authors thank the Agency for the Improvement of Higher Education Personnel, the National Council for Scientific Development and Technology, the Foundation for Research Support of Minas Gerais and Federal Institute of Minas Gerais—Campus Bambuí.

**Conflicts of Interest:** The authors declare that there is no conflict of interest.

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
