# Peer review of "Corn Cropping System and Nitrogen Fertilizers Technologies Affect Ammonia Volatilization in Brazilian Tropical Soils"

_soilsystems, doi:10.3390/soilsystems7020054_

Round 1

Reviewer 1 Report

          The authors' studies showed that the urea technologies reduce the N-NH3 losses compared to PU in both the examined NT and T systems, the losses under NT are higher than in the conventional system. Urea treated with 484 NBPT (1200 mg kg-1) alternative technology for efficient N utilization in grain production systems under NT as it has a 5-day delay on the day of maximum loss compared to. Also ammonium nitrate and sulphate represent a reasonable alternative to reduce N-NH3 losses in grain production systems. In the present study, a reduction in N-NH3 losses does not directly reflect an increase in yield and N uptake by maize. Probably because the ground buffers and levels very strongly. Therefore I would like to advise in chapter: 3.2. Effects of Fertilizers and Soil Nitrogen Stocks on Nutrient Accumulation and Corn Yield to include a table in which the nitrogen balance of the systems is compared and the reader can thus get an idea of the N efficiency. In this table, the N supply via fertilizer should be compared with the N removal via crops (maize). The difference between input and output is the balance as a variable for assessing the N efficiency. The N losses, measured as N-NH3 losses, can be subtracted from this balance. The resulting size can then be interpreted. Depending on the size, something can be estimated about the N supply from the soil or about the whereabouts of the nitrogen in a possibly positive balance.   The methodology is explained very extensively in all cases. Statistics, weather conditions, etc. Everything is very easy to understand and clearly presented. From my point of view, the measuring method for determining the N-NH3 losses could be explained very briefly as a principle. In practice, this consists in the neutralization of ammonia with phosphoric acid: 3 NH 3 + H 3 PO 4 ⟶ ( NH 4 ) 3 PO 4 to the PVC collectors adapted by Lara-Cabezas et al. Please briefly explain the measuring principle!   All in all, the work confirms already known theories and measurements. The work is technically very solid and resilient and creates a real basis for specific work in a specific region. It is also an important building block for the economy of agricultural production practice, the efficient use of resources and the avoidance of environmental pollution. My congratulations to the authors. A real advance for science and agriculture production!

Author Response

Dear reviewer;

We appreciate the opportunity to address the  suggestions/questions, which allowed us to clarify points in the manuscript that could lead to misinterpretation of the results. Points taken into consideration on the lines 199 to 204, 390 to 410, and 520 to 536.

Reviewer 2 Report

The manuscript was well written, minor revision is necessary.

1. Please add general conclusion in abstract

2. In Material and method, please specify the replication and the rate of each treatment as well as spacing of the corn planting 

3. In Result, please explain the accumulated ammonia loss with time; please refer the data to presented table or figure

4. In conclusion, please add general implication of the study

5. Please see the reviewed manuscript for revision

Author Response

Dear reviwer;

All comments were taken into account.

Reviewer 3 Report

Necessary changes and corrections are marked on the tex

Necessary changes and corrections are marked on the tex

Author Response

Dear reviewer;

We appreciate your contribuition to our manuscript. All comments were taken into account.
